# RNA-Seq Analysis of UPM-Exposed Epithelium Co-Cultivated with Macrophages and Dendritic Cells in Obstructive Lung Diseases

**DOI:** 10.3390/ijms23169125

**Published:** 2022-08-15

**Authors:** Paulina Misiukiewicz-Stępien, Michał Mierzejewski, Elwira Zajusz-Zubek, Krzysztof Goryca, Dorota Adamska, Michał Szeląg, Rafał Krenke, Magdalena Paplińska-Goryca

**Affiliations:** 1Department of Internal Medicine, Pulmonary Diseases and Allergy, Medical University of Warsaw, 02-091 Warsaw, Poland; 2Postgraduate School of Molecular Medicine, Medical University of Warsaw, 02-091 Warsaw, Poland; 3Department of Air Protection, Faculty of Energy and Environmental Engineering, Silesian University of Technology, 44-100 Gliwice, Poland; 4Genomic Core Facility, Centre of New Technologies, University of Warsaw, 02-097 Warsaw, Poland

**Keywords:** epithelium, urban particulate matter, UPM, asthma, COPD

## Abstract

Background. Elevated concentrations of airborne pollutants are correlated with an enlarged rate of obstructive lung disease morbidity as well as acute disease exacerbations. This study aimed to analyze the epithelium mRNA profile in response to airborne particulate matter in the control, asthma, and COPD groups. Results. A triple co-culture of nasal epithelium, monocyte-derived macrophages, and monocyte-derived dendritic cells obtained from the controls, asthma, and COPD were exposed to urban particulate matter (UPM) for 24 h. RNA-Seq analysis found differences in seven (CYP1B1, CYP1B1-AS1, NCF1, ME1, LINC02029, BPIFA2, EEF1A2), five (CYP1B1, ARC, ENPEP, RASD1, CYP1B1-AS1), and six (CYP1B1, CYP1B1-AS1, IRF4, ATP1B2, TIPARP, CCL22) differentially expressed genes between UPM exposed and unexposed triple co-cultured epithelium in the control, asthma, and COPD groups, respectively. PCR analysis showed that mRNA expression of BPIFA2 and ENPEP was upregulated in both asthma and COPD, while the expression of CYP1B1-AS1 and TIPARP was increased in the epithelium from COPD patients only. Biological processes changed in UPM exposed triple co-cultured epithelium were associated with epidermis development and epidermal cell differentiation in asthma and with response to toxic substances in COPD. Conclusions. The biochemical processes associated with pathophysiology of asthma and COPD impairs the airway epithelial response to UPM.

## 1. Introduction

Ambient air pollution is one of the key issues of public health. According to the WHO, one-in-nine deaths globally is caused by harmful air pollutants emitted from anthropogenic sources [1]. Airborne pollution is a mixture consisting of several gaseous components and a load of particulate matter (PM). The PM fractions are described according to an aerodynamic diameter (AD) of particles—AD ≤ 10 µm, AD ≤ 2.5 µm, and AD ≤ 0.1 µm, respectively, referred to as PM_10_, PM_2.5_, and UFP (ultrafine particles) [2]. Airborne pollution in highly populated areas is characterized by an increased content of fine particles originating from vehicle exhaust [3] and the coarse fraction emitted from household heating systems, particularly during the heating season [4]. Urban particulate matter (UPM) is a mixture of liquid and solid particles, which contains carcinogenic chemicals including inorganic ions, heavy metals, and polycyclic aromatic hydrocarbons (PAHs) characteristic for local urban, industry, and household errands [5]. The physical and chemical diversity of the inhaled compounds included in airborne pollution as well as the complex interplay and multiple interactions occurring within the airway epithelium result in various harmful effects.

Characteristics of airway epithelium including tightly clustered cells, a diversity of cell types with a large number of ciliated and mucus secreting cells as well as the production of a surfactant film contribute to human body protection not only against air pollutants, but also against different respiratory pathogens: viruses and bacteria. The respiratory epithelium actively regulates the local inflammatory response through interactions with the macrophages and dendritic cells (DCs). In our previous study, we found that active interactions between the epithelial cells and DCs are important components for the proper response of airways for air pollution exposure [6]. The in vitro effect of ambient air pollutants has been a subject of intense investigation. The exposure to air pollution resulted in the decreased antimicrobial properties of the airway epithelium and impacted the innate immunity [7]. Other effects of ambient air pollution on the respiratory epithelium were found to be associated with its toxicity [8], oxidative stress [9], and inflammation [10].

Pathophysiology of asthma and chronic obstructive pulmonary disease (COPD) is associated with respiratory epithelium dysfunction and destruction. Ongoing inflammation and remodeling significantly affect the airway microenvironment in asthma and COPD [11]. The metabolic reprogramming as well as the structural changes associated with these diseases modulate the epithelial response to airborne pollutants. Due to the loss of proper epithelium function and barrier integrity disruption, asthma and COPD patients are more susceptible to hazardous outcomes of airborne particulate matter. Elevated levels of harmful substances in the air are correlated with the increased prevalence of asthma and COPD as well as acute disease exacerbations [12,13]. However, detailed knowledge of the impact of the pathophysiology of obstructive lung diseases on the epithelial response to UPM exposure remains elusive. Therefore, we undertook a study aimed to evaluate the airway epithelium mRNA profile in response to airborne particulate matter. An in vitro triple cell co-culture model based on cells derived from healthy subjects and obstructive lung disease patients was used in this study.

## 2. Results

### 2.1. RNA-Seq Data Analysis

The RNA-Seq data analysis showed 68 differentially expressed genes (DEGs) between the mono- and triple co-cultures in the control group, 1396 DEGs between the mono- and triple co-cultures in asthma, and only three DEGs between the mono- and triple co-cultures in the COPD group (according to adjusted *p*-value *p* < 0.05). UPM exposure resulted in seven (CYP1B1, CYP1B1-AS1, NCF1, ME1, LINC02029, BPIFA2, EEF1A2), five (CYP1B1, ARC, ENPEP, RASD1, CYP1B1-AS1), and six (CYP1B1, CYP1B1-AS1, IRF4, ATP1B2, TIPARP, CCL22) DEGs between the UPM exposed and unexposed triple co-cultures in the control, asthma, and COPD groups, respectively (according to adjusted *p*-value *p* < 0.05) (Figure 1A). Considering the triple co-cultures exposed to UPM, we observed 5256 DEGs between the asthma and control group, 2297 DEGs between the COPD and control group, and 7591 DEGs between the COPD and asthma group (Figure 1B).

Volcano plots were used to visualize the detailed analysis of DEGs between the epithelium from the triple co-cultures exposed to UPM from the COPD and asthma group as well as between those groups and the control group separately (Figure 2A–C).

### 2.2. GO and KEGG Pathway Enrichment Analysis

The GO annotation revealed significantly changed (up- and downregulated) genes associated with the carbohydrate biosynthetic process (GO:0016051) in a comparison between the UPM stimulated and unstimulated epithelium from a triple co-culture of the controls. Several terms were also strongly related to lipid biosynthesis and metabolism. The differentiating genes after UPM stimulation in the control epithelium from the triple co-cultures (Figure 3A) were associated with the sterol biosynthetic process (GO:0016126), cholesterol biosynthetic process (GO:0006695), steroid biosynthetic process (GO:0006694), and steroid metabolic process (GO:0008202).

The GO analysis of genes differentiating the UPM exposed epithelium from triple co-culture in the asthma group and corresponding unexposed epithelium co-culture revealed an association with epidermis development (GO:0008544) and epidermal cell differentiation (GO:0009913) with 35 and 26 DEGs, respectively. Moreover, this comparison also demonstrated a significant enrichment in genes associated with the humoral immune response (GO:0006959) and granulocyte chemotaxis (GO:0071621) (Figure 3A).

Genes differentiating the UPM-exposed COPD epithelium (triple co-cultured) from the corresponding unstimulated one were associated with response to toxic substance (GO: 0009636) with 40 DEGs. Furthermore, the GO analysis in this comparison identified up- and downregulated genes assigned to terms involved in the migration of leukocytes (GO:0050900), granulocytes (GO:0097530), neutrophils (GO:1990266), and the reactive oxygen species metabolic process (GO:0072593) (Figure 3A).

Comparison between the asthmatics and controls in terms of the GO analysis of the UPM exposed epithelium from triple co-cultures revealed up- and downregulated genes associated with ribonucleoprotein complex biogenesis (GO:0022613), ncRNA processing (GO:0034470), and mitochondrial translational termination (GO:0070126) (Figure 3B), while a comparison between the control and COPD (triple co-cultured epithelium) showed GO terms associated with mitochondrial translational elongation (GO:0070125) and termination (GO:0070126), oxidative phosphorylation (GO:0006119), mitochondrial gene expression (GO:0140053), and cellular protein complex disassembly (GO:0043624) (Figure 3B).

In addition, the KEGG pathway analysis revealed six significantly overrepresented pathways in the controls after UPM stimulation (Figure 4A). Significant associations with rheumatoid arthritis and viral protein interaction with cytokine and cytokine receptors among genes differentiating the UPM stimulated asthmatic epithelium compared to the unstimulated one were observed (Figure 4B). In the COPD group, UPM exposure resulted in the enrichment of pathways especially associated with steroid hormone biosynthesis, cytokine–cytokine receptor interaction, and viral protein interaction with cytokine and cytokine receptors compared to the unstimulated group (Figure 4C). Additionally, we observed 16 enriched pathways in the COPD group after UPM exposure compared to the control group (Figure 4D). No significant differences after UPM stimulation were established in the KEGG analysis between the epithelium from the asthma and control groups as well as between the COPD and asthma groups.

### 2.3. RT-qPCR Analysis

The genes selected from RNA-Seq analysis with a significant and the highest fold change of mRNA expression were chosen for qRT-PCR verification (Figure 5A,B). Genes with *p*-value (corrected) lower than 0.1 in comparison between the UPM exposed and nonexposed epithelial cells from monoculture or triple co-culture in the control, asthma, and COPD groups were selected for verification. Differences in expression observed in RNA-Seq for the selected genes are illustrated in Figure 5 as follows:
(a)Fold change of mRNA expression between the UPM exposed and nonexposed epithelial cells from the triple co-culture (Figure 5A, *Y*-axis) was compared to the fold change of mRNA expression between the UPM and no UPM exposed epithelial cells from the monoculture (Figure 5A, *X*-axis). A separate plot (and gene selection) was prepared for each group (control/asthma/COPD).(b)Fold change of mRNA expression between the UPM exposed and nonexposed epithelial cells from the triple co-culture in one of the clinical group was plotted against the same value in other clinical groups (three panels: asthma–control, COPD–control, COPD–asthma).

The detailed list of 17 tested candidate genes (AHRR, ARC, ATP1B2, BPIFA2, CCL22, CYP1B1, CYP1B1-AS1, EDC3, EEF1A2, ENPEP, LINC02029, IRF4, ME1, NCF1, RASD1, RMDN2-AS1, TIPARP) is described in Appendix A, Table A1. The results obtained in qRT-PCR were close to these obtained by the RNA-Seq measurements. In most cases, UPM exposure upregulated gene expression or remained unchanged except for CCL22 and RMDN2-AS1 for all groups and CYP1B1 and LINC02029 for the controls and asthma, which were suppressed in almost all tested combinations. qRT-PCR confirmed several significant changes in the mRNA expression in the epithelium after UPM exposure. The results of the PCR analysis showed that the expression of the evaluated mediators in the epithelial cells co-cultivated with moMφs and moDCs was much stronger than in the epithelial cells cultured alone. The overall *p*-values of the selected genes’ mRNA expression comparisons between the control, asthma, and COPD groups in the UPM exposed epithelium from mono- and triple co-cultures are presented in Appendix A, Table A2. PCR analysis highlighted the differences in the epithelial response after UPM stimulation between the controls and patients with obstructive lung diseases. We found that the mRNA expression of BPIFA2 and ENPEP was upregulated after UPM treatment, most potently in triple co-cultures of the asthma and COPD patients only. Additionally, it seems that the epithelial cells from COPD patients highly activated CYP1B1-AS1, ME1, TIPARP mRNA expression after UPM stimulation in contrast to the control and asthma group, where these changes were not observed (Figure 6).

## 3. Discussion

Obstructive lung diseases carry a particular risk of serious consequences of air pollution exposure. Our study, which used an advanced triple cell co-culture model, showed a distinct pattern of transcriptomic changes after UPM exposure in the epithelium from triple co-cultures in healthy controls, asthma, and COPD patients. Pre-existing obstructive lung diseases were associated with considerable changes in gene expression in the UPM-exposed epithelium, as we observed 5256 DEGs in a comparison between the asthma and control group, 2297 DEGs between the COPD and control group, and 7591 DEGs between the COPD and asthma group. Our results revealed the genes and biological processes apparently involved in response to UPM exposure. We showed that the most potently activated genes after air pollution exposure in the triple co-cultured epithelium of asthma and COPD patients were BPIFA2 and ENPEP, while CYP1B1-AS1 and TIPARP were upregulated in the COPD epithelium only. Here, for the first time, we present a comprehensive characterization of the molecular processes taking place in the respiratory epithelium after UPM stimulation in the in vitro model, which considers the impact of the interaction of the epithelial–macrophage–dendritic cells in the healthy controls, asthma, and COPD patients.

The effect of airborne PM on the monocultured airway epithelium has been well-characterized. The exposure of the ALI cultured epithelium resulted in toxic effects associated with oxidative stress, pro-inflammatory response as well as enhanced cytotoxicity [14]. Our study, which showed the upregulation of genes associated with response to toxic substances, is in line with those earlier observations. CYP1B1-AS1 is a member of the long-noncoding antisense RNAs located on chromosome 2 [15] and is considered to be a positive regulator (i.e., enhancer) of CYP1B1 transcription [16]. CYP1B1 belongs to the cytochrome family and is linked to xenobiotic metabolism. Our study demonstrated that the epithelium of COPD patients was more susceptible to UPM exposure than the asthmatic as well as control epithelium and showed the highest expression of CYP1B1-AS1 in both the mono- and triple co-cultures. Earlier studies showed the upregulation of CYP1B1-AS1 expression in oral masticatory mucosa from cigarette smokers compared to never smokers [17] and in A549 cells exposed to cigarette smoke extract [18]. The expression measured in our study was also significantly elevated in the UPM-exposed COPD epithelium cultures. As the COPD patients included in our study were active smokers, it can be supposed that epithelium pre-exposure to cigarette smoke manifests as increased vulnerability to UPM stimulation. Bioinformatic analysis of our RNA-Seq results revealed significant upregulation of epithelial CYP1B1 expression within all tested groups after 24 h of UPM exposure but PCR revealed an insignificant elevation of CYP1B1 expression in the COPD group only. CYP1B1 as well as CYP1A1 are aryl hydrocarbon receptor-dependent (AHR) and are considered as the markers of AHR activation [19]. AHR is involved in xenobiotic metabolism, especially aromatic hydrocarbons (PAHs), compounds embedded on airborne PM [20]. AHR can be repressed by the aryl hydrocarbon receptor repressor (AHRR). Previous study showed that wood smoke, urban fine particulate matter, and PAHs increased the AHRR expression in the airway epithelial cells [21]. Our data showed that AHRR expression tended to be decreased in asthma while it was increased in the COPD triple co-cultured epithelium after UPM exposure. Cell AHR signaling is mediated by TCDD-inducible poly-ADP-ribose polymerase (TIPARP) expression [22], or by TIPARP via ADP-ribosylation [23]. Increased expression of TIPARP was shown to decrease not only AHR activity but also AHR-associated genes, suggesting its role in the negative regulation of the AHR-pathway [24]. Our results showed that upregulation of TIPRAP mRNA expression after UPM stimulation in the epithelium from COPD patients. As in previous murine studies, the downregulation of TIPARP expression increased the sensitivity to TCDD-dependent toxicity [25], and we suggest that increased TIPARP expression in the UPM-exposed COPD epithelium is associated with enlarged UPM toxicity within this group. Our study may imply that air pollution highly elevates oxidative stress in the airway epithelium, especially for COPD patients, which can cause disease exacerbation or other pathological processes such as carcinogenesis through the downregulation of protective defense biochemical mechanisms of the airway epithelium. Thus, combining the results of the current study and a common knowledge of the impairment of antioxidant defense in COPD, a preventive antioxidant treatment for COPD patients susceptible to air pollution seems to be a reasonable approach.

Our study revealed an increased expression of ENPEP in the asthma and COPD epithelium triple co-cultures exposed to UPM. Glutamyl aminopeptidase (ENPEP) is a mammalian type II integral membrane zinc-containing endopeptidase belonging to the aminopeptidase family. ENPEP is a crucial regulatory factor of blood pressure, taking part in blood vessel remodeling [26]. The exact role of ENPEP in respiratory physiology is not explained yet, but recent studies suggest a strong correlation between ENPEP and angiotensin converting enzyme 2 (ACE2) expression [27]. The possible mediatory role of ENPEP in UPM-induced epithelial changes in patients with obstructive respiratory diseases is still a novel aspect to elucidate. We can only speculate that due to its aminopeptidase activity that this marker might be associated with cell activation, signal transduction, and cell-matrix adhesion in the asthmatic and COPD airways after air pollution exposure.

Another important mediator of UPM induced epithelial response revealed by our study is BPIFA2, a member of the palate, lung, and nasal epithelium clone (PLUNC) protein family, encoded by the gene cluster located on chromosome 20 [28]. Proteins included in this family are associated with local antibacterial responses in the nose, mouth, and upper respiratory tract [29]. Kang et al. found an upregulation of BPIFA2 after PM_10_ exposure in several cell lines such as normal lung epithelial cells (BEAS-2B), human lung carcinoma (A549), and human bronchiolar carcinoma (NCI-H358) [30]. The activation of BPIFA2 after UPM exposure in asthma and COPD epithelium from the triple co-cultures in our study suggest the upregulation of innate immune response in the respiratory tract of patients with obstructive lung diseases after air pollution exposure.

The GO terms and KEGG pathways analysis in our study demonstrated that biological processes in epithelial cells from patients with obstructive lung diseases exposed to UPM differed considerably compared to the healthy individuals. In healthy subjects, UPM treatment altered pathways associated with lipid metabolism and glucose catabolism. Similar effects have already been reported in several previous studies [31,32]. Our results showed upregulated NADH regeneration and glucose catabolic processes, which reflects a preserved regenerative capacity of healthy epithelium. In contrast, the exposure to hazardous airborne material such as cigarette smoke was shown to downregulate glucose metabolism and increase fatty acid oxidation (FAO) with simultaneous enzymatic pathway alterations in the lung alveolar cells [33]. These processes caused cellular damage and surfactant deficiency, leading to impaired lung function in the smokers and COPD patients [34]. Likewise, in our study, the UPM-exposed COPD epithelium from triple co-cultures showed the upregulation of pathways associated with response to toxic substances, mitochondrial associated pathways, oxidative stress as well as ROS metabolism. Interestingly, Leclercq et al. found that despite an enhanced expression of genes involved in the metabolism of harmful substances, the COPD epithelium showed a decreased response capacity to air-pollution-derived hazardous compounds [35]. Other authors have reported that free radicals formed in cells resulted in expanded oxidative stress, causing the increased transcription of pro-inflammatory genes via the NF-kB pathway such as IL-8 as well as epigenetic changes such as histone acetylation, further increasing DNA unfolding and the transcription of pro-inflammatory genes [36]. Furthermore, COPD patients revealed increased PM_10_-induced genotoxicity compared to the healthy subjects [37]. Importantly, our results found that UPM exposure of the COPD epithelium also resulted in the increased expression of genes associated with the migration of granulocytes, especially neutrophils. It is known that the long-lasting presence and activity of neutrophils in the airways leads to the release of cytotoxic and profibrotic agents, resulting in local tissue remodeling by injury and fibrosis. We also observed the deregulation of genes associated with cellular protein complex disassembly in the UPM-exposed COPD epithelium compared with the healthy one. It has been reported that the exposure to airborne pollutants induce methionine oxidation, resulting in protein-misfolding and endoplasmic reticulum stress in chronic lung diseases [38,39]. We suggest that UPM-induced protein reorganization is associated with highly activated oxidative stress, and disturbed COPD epithelium layer structure and function, which may accelerate the COPD exacerbations induced by air pollution.

In contrast to COPD, the UPM-exposed asthma epithelium was characterized by a deregulation of distinct pathways, especially biological processes associated with humoral immune response and granulocyte chemotaxis. Our results are consistent with other studies. Using a murine asthma model, Huang et al. showed that the exposure to airborne PM supported the intensified neutrophil recruitment and induction of Th1-related cytokine synthesis (TNF-α and IFN-γ) and resulted in allergic-like immune responses including increased eosinophil influx and upregulated Th2-cell mediated cytokine production (IL-5 and IL-13) [40]. Eosinophils are important sources of transforming growth factor β (TGF-β), which mediates the induction of structural changes in asthmatic airways associated with subepithelial fibrosis, myocyte hyperplasia, and hypertrophy, disruption of epithelial integrity, goblet cell metaplasia, and vascular permeability [41]. Our results also showed a deregulated pathway associated with the retinoid metabolic process in the UPM-exposed epithelium. It has been demonstrated that long-lasting interactions of retinoids with their overexpressed receptors on asthmatic bronchial epithelium enable an aberrant tissue repair and rebuilding via TGF-β1 synthesis [42]. Moreover, results of our study showed UPM dependent regulation of epidermis development and epidermal cell differentiation pathways. These biological processes contribute to the deregulation of epithelial tight junction integrity, wound healing, tissue repair, and by stimulating cell proliferation [28]. Our study revealed that the main biological processes activated by air pollution exposure in asthmatic airways are associated with inflammation, especially granulocyte chemotaxis, humoral immune response, and the disruption of epithelial integrity, which as a consequence result in the loss of cellular defense mechanisms.

Our study had some limitations. First, a small but statistically optimized group of healthy subjects, asthmatics, and COPD patients was included in the study. Verification of RNA-Seq results by the qRT-PCR measurements was performed in extended groups. Second, our model contained nasal epithelium, a non-invasively obtained functional substitute of bronchial epithelial cells. Our unique cell co-culture model certainly did not mimic all of the pathophysiological processes involved in obstructive lung diseases, but it seems useful to determine the impact of cell–cell interactions on the epithelium transcriptome after exposure to UPM between healthy subjects and patients with asthma and COPD. Third, our results were obtained by a full transcriptome analysis using RNA-Seq. Although this method did not allow us to provide detailed information concerning the UPM impact on specific epithelium cell types constituting the epithelium layer, it let us recognize the wide picture of transcriptome differences caused by UPM exposure among all of the tested groups. As the epithelium layer consists of several cell types, further investigation in this field of study using single-cell RNA (sc-RNA) sequencing might be a promising approach to reduce the information noise associated with epithelial cell heterogeneity.

## 4. Materials and Methods

### 4.1. Patient Characteristics

The study involved 10 asthma patients, eight patients with COPD, and eight healthy subjects. The diagnosis of asthma and COPD was established according to the current GINA and GOLD reports, respectively [43,44]. The control group consisted of healthy individuals with no airway obstruction confirmed by normal spirometry results. The exclusion criteria were as follows: treatment with systemic or nasal steroids, asthma, or COPD exacerbation within 3 months from sampling, and symptoms of respiratory tract infection in the preceding 3 months. Peripheral blood samples and nasal brushing were collected from each participant. RNA-Seq analysis was performed in a group of 12 subjects (four controls, four asthma, four COPD) (Table 1). The COPD patients were heavy smokers, in contrast to the asthma patients and healthy controls. The clinical characteristics of all patients and controls recruited to the study are summarized in Appendix A, Table A3.

The study protocol was approved by the Ethics Committee of the Medical University of Warsaw (KB/37/2020) and written informed consent was obtained from all participants.

### 4.2. Flow Cytometry Analysis

Cells were stained with antibodies against the surface binding molecules CD45 (APC-H7), CD326 (PerCP-Cy5-5), MUC1 (BV605) (BD Biosciences, San Jose, CA, USA), and incubated for 20 min in the dark at RT. After washing away the reagents, the cells were fixed and permeabilizated using lysis buffer and permeabilization solution 2 (BD Biosciences, San Jose, CA, USA), then stained with intracellular markers β-tubulin (Alexa fluor 488) and cytokeratin (BV510) (BD Biosciences, San Jose, CA, USA) in BD Horizon Brilliant Stain Buffer (BD Biosciences, San Jose, CA, USA) for 20 min in the dark. Cells were analyzed by flow cytometry using the FACS Celesta instrument (BD Biosciences, San Jose, CA, USA) equipped with blue (488-nm), violet (405-nm), and red (640-nm) lasers. Unstained cells and compensation beads (BD Biosciences, San Jose, CA, USA) were used to set voltages and create single stain negative and positive controls. Compensation was set to account for the spectral overlap between the seven fluorescent channels used in the study. Cells with a basal phenotype were identified as CD45-CD326 + cytokeratin+, with the secretory phenotype as CD45-CD326 + MUC1+, with the ciliated phenotype as CD45-CD326 + β-tubulin+.

### 4.3. Hematoxylin and Eosin (H&E) Staining

Cultures were fixed with 10% neutral buffered formalin, pre-embedded in the desired orientation in premelted 1% agarose (Sigma Aldrich, St. Louis, MO, USA), processed on paraffin blocks with the standard protocol, cut into 5 µm sections, and mounted on positively charged glass slides (Leica, Wetzlar, Germany). Slides were dried in 60 °C for 30 min for paraffin melting and further dewaxed and hydrated with xylene and decreasing concentrations of alcohols. Sections were then stained with hematoxylin and eosin (H&E) and analyzed under light microscope equipped with a digital camera PrimoStar with AxioCam ERc5s (Zeiss, Oberkochen, Germany).

### 4.4. Cell Culture and Scheme of the Study

The nasal epithelial cells were isolated, cultivated, and specialized as previously described [45]. In brief, nasal epithelial cells were isolated from brush swabs (Cytobrush Plus GT, CooperSurgical, San Ramon, CA, USA) sampled at the interior surface of both nostrils. Then, the cells were cultured in the air–liquid interface (ALI) for 21 days in PneumaCult-ALI medium (StemCell, Vancouver, BC, Canada). Macrophages and DCs were isolated from a peripheral blood sample by Lymphoprep (StemCell, Vancouver, BC, Canada) centrifugation. The PBMC were frozen and thawed for specialization. Monocyte derived macrophages (moMφs) were specialized by 20 ng/mL M-CSF (StemCell, Vancouver, BC, Canada) stimulation for 10 days and monocyte derived DCs (moDCs) by cultivation in combination with 40 ng/mL GM-CSF (StemCell, Vancouver, BC, Canada), 20 ng/mL IL-4 (StemCell, Vancouver, BC, Canada) for 8 days, 50 ng/mL TNF-α (Bio-Techne, Minneapolis, MN, USA), and 50 ng/mL IL-1β (StemCell, Vancouver, BC, Canada) on the sixth day of specialization.

Fully differentiated ALI cultures of epithelial cells were cultured in a two-chamber system (Greiner Bio-One, Kremsmünster, Austria) (Figure 7 and Figure 8). The triple co-cultures were prepared as previously described [46]. Briefly, the triple-co-cultures contained nasal epithelial cells cultured in ALI conditions for 21 days, fully differentiated moMφs (cultured for 10 days before the experiment) located on the top of the epithelium, and subepithelial moDCs (in eighth day of their specialization). moMφs and moDCs in triple co-cultures were suspended in media without supplements. The co-cultures were autologous (i.e., for each co-culture, the epithelial cells, macrophages and moDC were obtained from the same individual). UPM was added to the epithelial cells after the cells were combined in a triple-co-culture. Epithelial cells were cultured with or without stimulation with 100 µg/mL UPM (10 μL of UPM stock solution added on the top of the epithelial cells) for 24 h in a scheme as follows:(1)Epithelial cells (monoculture);(2)Epithelial cells + moMφs + moDCs (triple co-culture).

The epithelial cells cultured in ALI conditions contained a domination of cells with the ciliated and secretory phenotype, and less basal epithelial cells (Appendix A, Figure A1). The co-cultivation of epithelial cells with moMφs and moDCs did not change the proportion of epithelial subpopulations Appendix A Figure A2).

After 24 h, the cells were harvested, moMφs and moDCs were rinsed off, and the epithelium was used in the RNA-Seq analysis.

### 4.5. Particle Preparation

The filters with urban particulate matter were provided by the Silesian University of Technology. The samples were collected with a low-volume PM sampler type PNS-15 (Atmoservice, Poland) 1.5 m above the ground level, at a flow rate of 2.3 m^3^/h, according to the PN-EN 12341:2006 standard [47] in Zabrze, Gliwice, and Żory during the heating season as published previously [6]. These cities are located in the Upper Silesia Region, which, compared with other EU countries as well as other Polish regions, is characterized by relatively high levels of PM. Airborne particulate matter was collected on high-purity quartz (SiO_2_) microfiber filters (QM-A, Whatman, Little Chalfont, Buckinghamshire, UK). The heavy metal content in UPM was previously described [6]. The particles were detached from the filters by sonication and filtrated through strainers with 70 μm pores (Corning, Corning, NY, USA). The sediment of the particles was dried at 96 °C to dry mass, weighted, resuspended in PBS into stock solution containing 10 µg/µL UPM, and autoclaved.

### 4.6. RNA Isolation

After 24 h of incubation, the UPM DCs and macrophages were washed out. The attached cells were collected, the total RNA was isolated by the TRI reagent (Sigma-Aldrich, St. Louis, MA, USA) method, and further purified with NucleoSpin RNA (Machery&Nagel, Düren, Germany) using the protocol including DNA digestion. The concentration and quality of the isolated RNA were determined using Nanodrop 2000 (Thermo Fisher Scientific, Waltham, MA, USA) and validated by Agilent Bioanalyzer 2100 with an RNA 6000 Pico Kit (Agilent, Santa Clara, CA, USA). The control of possible mycoplasma contamination was analyzed using the MycoSPY Kit (Biontex, Mainz, Germany).

### 4.7. RNA-Seq Analysis

The mRNA sequencing was performed in four asthmatics, four COPD patients, and four control samples. Libraries for the RNA-Seq measurements were prepared according to the manufacturer’s protocol for the KAPA mRNA HyperPrep Kit (Roche, Basel, Switzerland). A total of 250 ng of intact total RNA was subjected to heat fragmentation (94 °C, 5 min or 85 °C, 6 min for partially degraded samples) and TruSeq Unique dual index adapters (Illumina, San Diego, CA, USA) were used. Twelve cycles of library amplification were applied. The size distribution of the final libraries were validated using Agilent Bioanalyzer 2100 and a High Sensitivity DNA Kit (Agilent, Santa Clara, CA, USA). The final library concentration was determined by qPCR using a Kapa Library Quantification Kit (Roche, Basel, Switzerland). Sequencing was performed using Illumina NovaSeq 6000 with the NovaSeq 6000 S1 Reagent Kit (Illumina, San Diego, CA, USA), generating 2 × 100 pair-end reads using the manufacturer’s standard protocols. High quality output data were obtained (more than 98% of data with quality exceeding Phred Score Q37) in an amount of 36–55 MR/sample.

### 4.8. Bioinformatic Analysis

Raw sequences were trimmed according to quality using Trimmomatic [48] (version 0.39) with default parameters, except MINLEN, which was set to 50. Trimmed sequences were mapped to the human reference genome provided by ENSEMBL, (version grch38_snp_tran) using Hisat2 [49] with default parameters. Optical duplicates were removed using the Mark Duplicates tool from the GATK [50] package (version 4.1.2.0) with default parameters, except with OPTICAL_DUPLICATE_PIXEL_DISTANCE set to 12,000. Reads that failed to map to the reference were extracted using Samtools [51] and mapped to the Silva meta-database of rRNA sequences [52] (version 119) with Sortmerna [53] (version 2.1b) using the “–best 1” option. Mapped reads were associated with transcripts from the GRCh38 database [54] (Ensembl, version 77) using HTSeq-count [55] (version 0.9.1) with the default parameters except with the stranded set to “reverse”. Differentially expressed genes were selected using the DESeq2 package [56] (version 1.16.1). Fold change was corrected using apeglm [57]. *p*-values were corrected for the multiple hypothesis test with the Benjamini–Hochberg algorithm. To provide equal power during testing overrepresentation of the Gene Ontology (GO) [58] terms and Kyoto Encyclopedia of Genes and Genomes (KEGG) [59] categories, the same proportion of genes was selected for each comparison. The top 5% of genes (according to *p*-value) were tested for overrepresentation vs. the whole set of genes with detectable expression. The assessment was carried out with cluster profiler package [60]. The RNA-Seq data were uploaded to GEO Omnibus (reference no. GSE175541). https://www.ncbi.nlm.nih.gov/geo/query/acc.cgi?acc=GSE175541, accessed on 6 August 2022.

### 4.9. qRT-PCR Measurements

cDNA synthesis was conducted using the High Capacity cDNA Reverse Transcription Kit with RNase Inhibitor (Thermo Fisher Scientific, USA). Quantitative real-time PCR was performed to assess the mRNA expression of AHRR, ARC, ATP1B2, BPIFA2, CCL22, CYP1B1, CYP1B1-AS1, EDC3, EEF1A2, ENPEP, IRF4, LINC02029, ME1, NCF1, RASD1, RMDN2-AS1, TIPARP, and 18s rRNA in the epithelial cells. The quantitative real-time PCR analysis was performed on an ABI-Prism 7500 Sequence Detector System (Applied Biosystems, Thermo Fisher Scientific, Waltham, MA, USA). The primer specification is shown in Appendix A, Table A4. Relative quantification values were calculated by the 2-∆∆CT method and 18s rRNA was applied for each sample as an internal control in order to normalize the gene expression levels. The unstimulated epithelial cells from monocultures of each individual were used as a calibrator.

### 4.10. Statistical Analysis

Statistical analysis was performed with the use of Statistica 13.3 software package (StatSoft Inc., Tulsa, Ok, USA), GraphPad (version 9.3.1 GraphPad Software, Inc., San Diego, CA, USA. https://www.graphpad.com/, accessed on 27 October 2021), or the R environment (version 4.1.0, https://cran.r-project.org/, accessed on 18 May 2021). The Kruskal–Wallis test, followed by the Dunn’s post hoc test, was used to assess the differences between the continuous variables in the three study groups. The Mann–Whitney U test was applied for pairwise comparisons. The Pearson Chi-square test was used to compare the inter-group differences between the categorical variables. The results are given as the median and interquartile range (IQR). Differences were considered statistically significant at *p* < 0.05.

## 5. Conclusions

The response of the asthma and COPD epithelium to UPM stimulation was distinct from the healthy subjects. Our study strongly suggests a cellular genetic reprogramming after UPM exposure in the asthma and COPD patients compared to the healthy subjects, which was associated with ongoing pathophysiological processes in obstructive lung diseases. Based on the presented results, we propose selected genes as potential markers of progressive epithelial cell activity and possible cell damage associated with UPM exposure in patients with asthma and COPD. Further research focused on the suggested target genes and affected biochemical processes might contribute to the development of novel therapeutic approaches for the effective treatment of UPM-associated exacerbations of obstructive lung diseases.

## Figures and Tables

**Figure 1 ijms-23-09125-f001:**
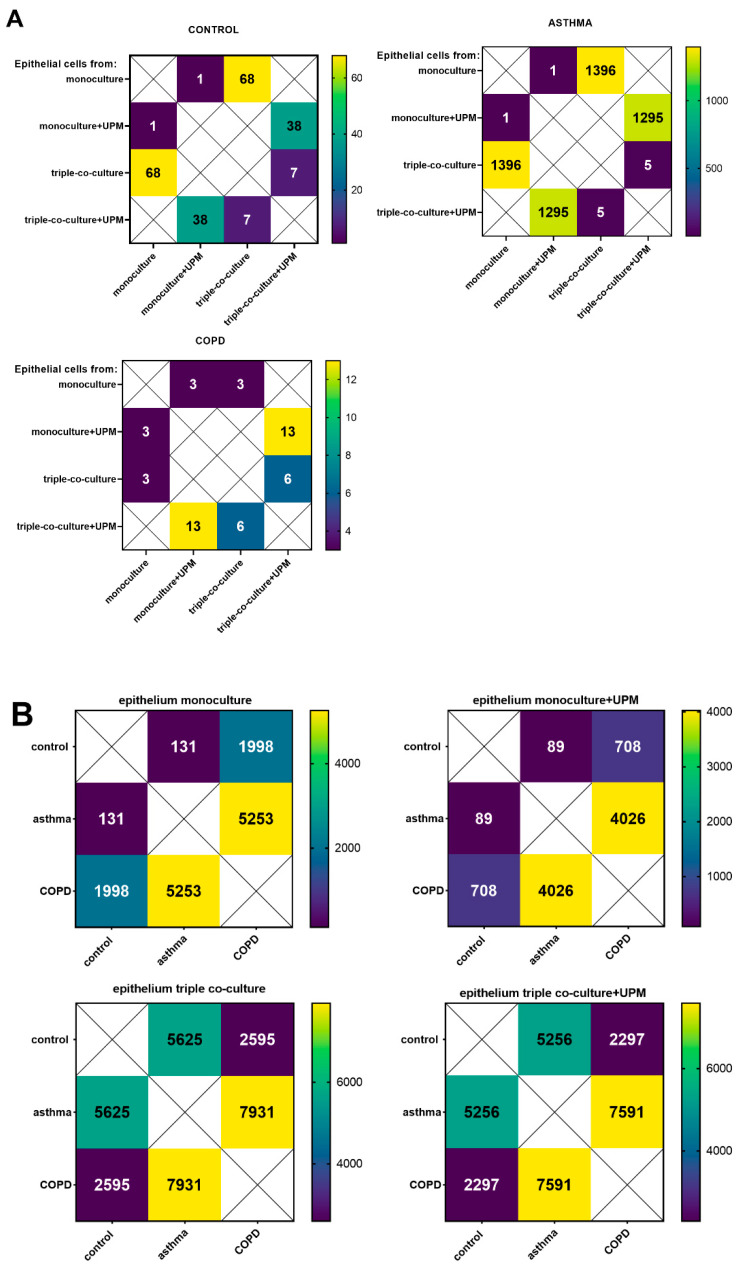
The heat map of the number of differentially expressed genes (DEGs) within the control, asthma, and COPD groups related to UPM exposure (**A**) and the DEGs between the control, asthma, and COPD groups in the same stimulation model (**B**). The values represent the number of DEGs (according to adjusted *p*-value, *p* < 0.05) in a comparison between the groups: epithelium from the monoculture, epithelium from the monoculture exposed to UPM for 24 h, epithelium from the triple co-culture, and epithelium from the triple co-culture exposed to UPM for 24 h, UPM—urban particulate matter.

**Figure 2 ijms-23-09125-f002:**
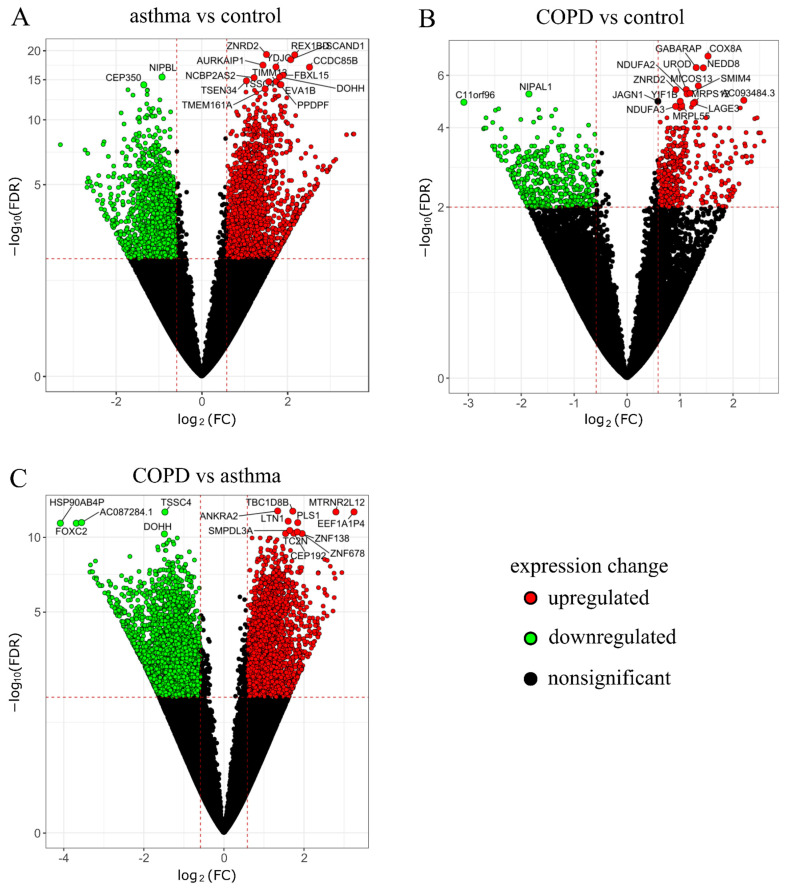
Volcano plots for the comparison between the UPM exposed epithelium (from triple co-cultures) in the asthma and control group (**A**), the COPD and control group (**B**), the COPD and asthma group (**C**). Genes are colored by fold change (FC). The *x*-axis illustrates the fold change (FC) (log-scaled) while the *y*-axis indicates the false discovery rate (FDR) adjusted *p*-values (log-scaled). Red points represent increased (upregulated) genes, green points stand for decreased (downregulated) genes, and black points show non-significantly deregulated genes.

**Figure 3 ijms-23-09125-f003:**
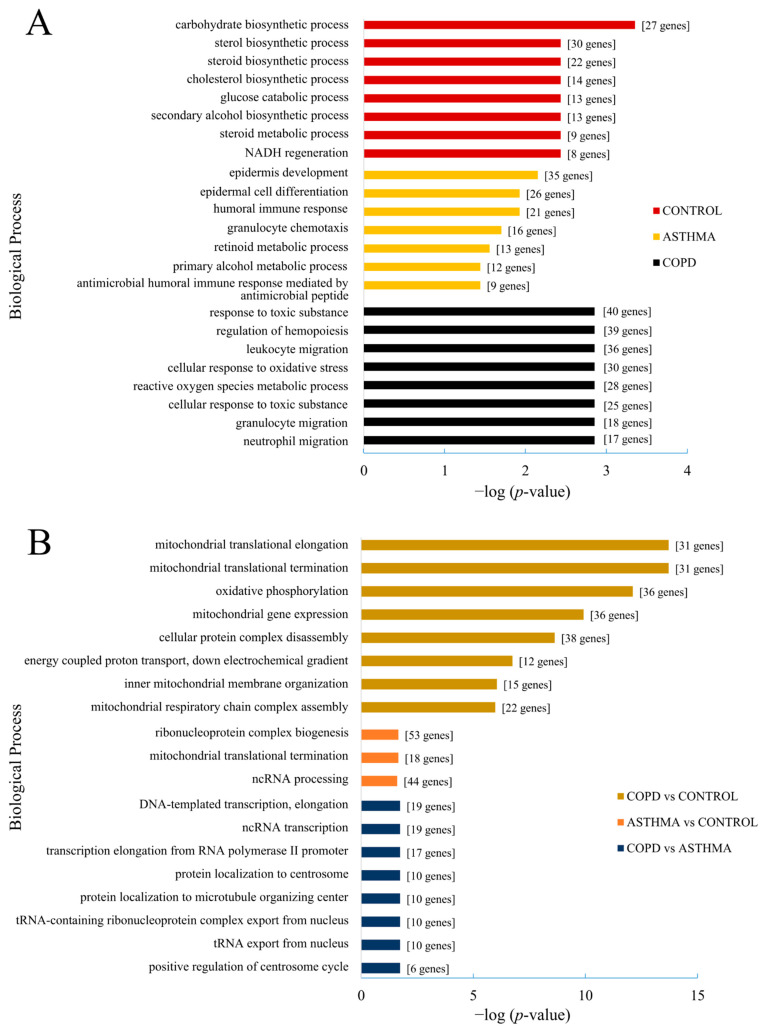
Gene Ontology (GO) analysis of the most significantly up- and downregulated terms (categories among the top 5% of genes) according to the biological process in the epithelium (from triple co-culture) after UPM exposure (**A**) and in a comparison between the control, asthma, and COPD groups (**B**). Analysis according to the adjusted *p*-value.

**Figure 4 ijms-23-09125-f004:**
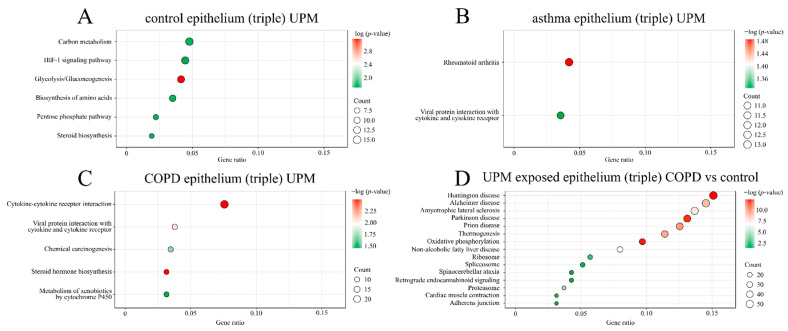
The Kyoto Encyclopedia of Genes and Genomes (KEGG) (categories among the top 5% of genes) significantly enriched pathways after UPM exposure in the control (**A**), asthma (**B**), and COPD (**C**) epithelium from the triple co-culture and in the UPM-exposed COPD epithelium compared to the control (**D**). Analysis according to the adjusted *p*-value.

**Figure 5 ijms-23-09125-f005:**
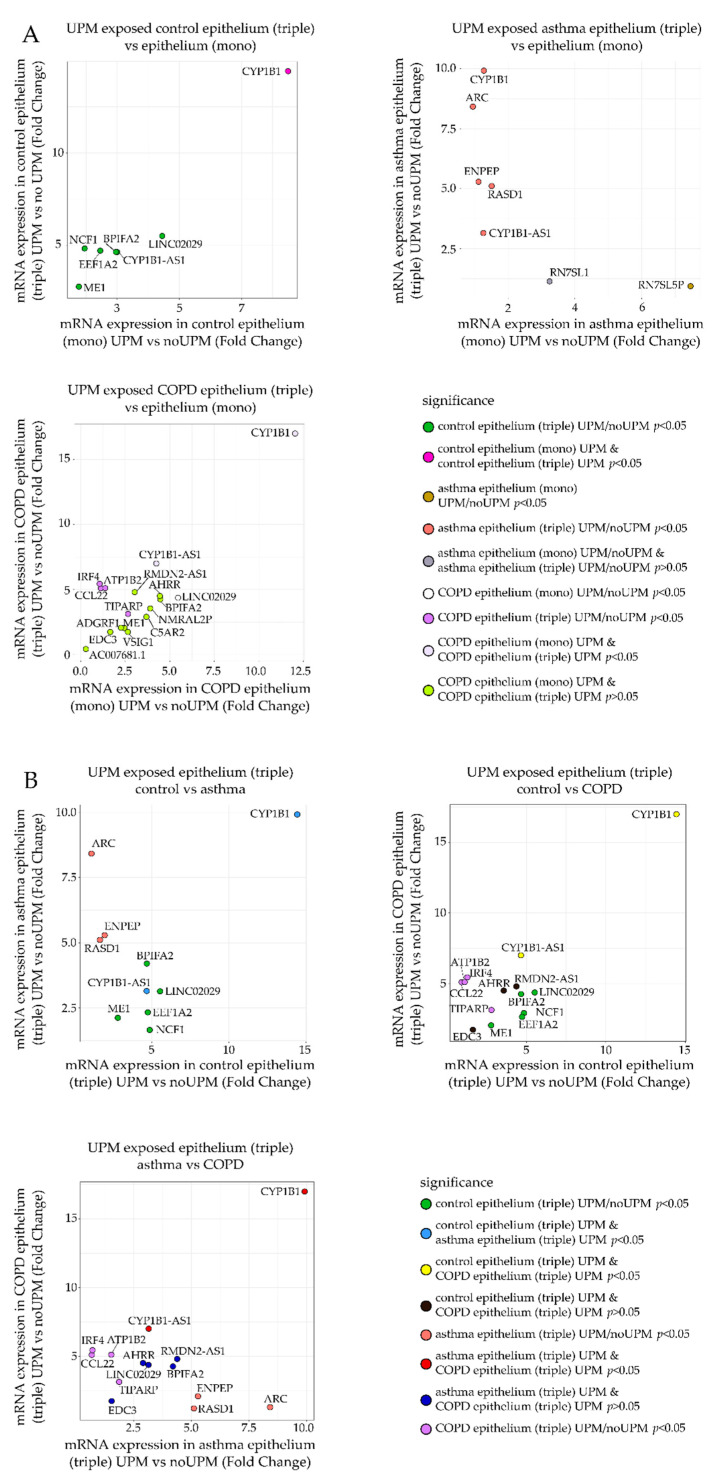
A comparison of the mRNA expression (fold change) (**A**) in the epithelium after UPM exposure from mono- and triple co-cultures and (**B**) between the UPM-exposed triple co-cultures in the control, asthma, and COPD groups.

**Figure 6 ijms-23-09125-f006:**
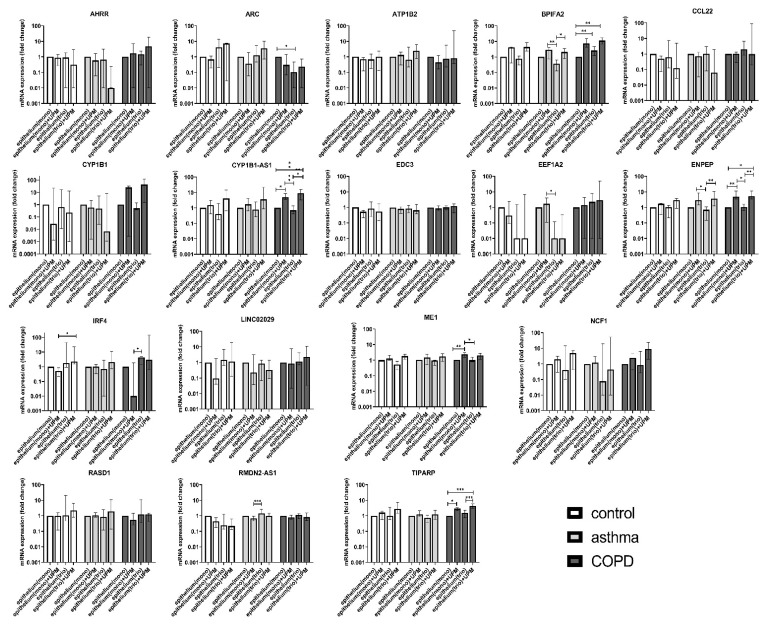
The mRNA expression of selected markers in the epithelium after 24 h of UPM exposure in the mono- and triple co-cultures in the control subjects, asthma, and COPD patients. The data are shown as interquartile range (whiskers), and median (column); *p*-value was calculated using the Kruskal–Wallis test followed by Dunn’s post hoc test. * *p* ≤ 0.05; ** *p* ≤ 0.01; *** *p* ≤ 0.001.

**Figure 7 ijms-23-09125-f007:**
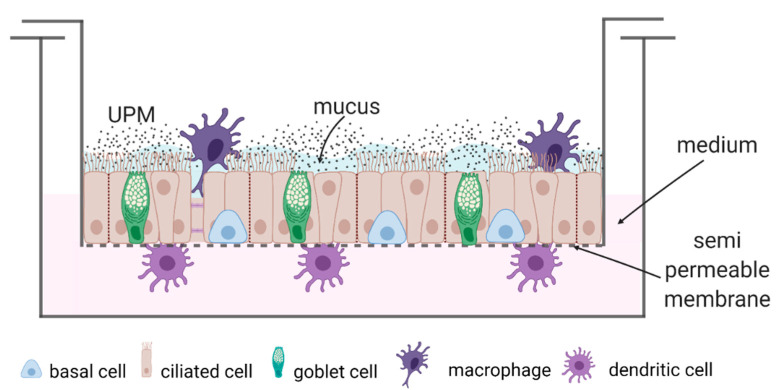
A schematic drawing of the triple co-culture stimulated with UPM.

**Figure 8 ijms-23-09125-f008:**
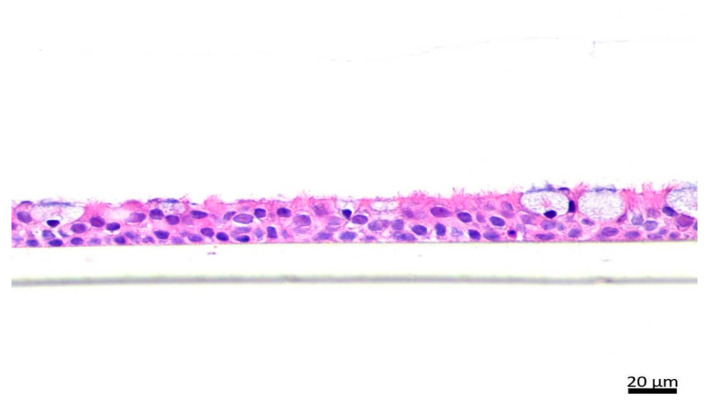
The hematoxylin and eosin (H&E) staining of the differentiated nasal epithelial cells cultured in ALI conditions for 21 days (original magnification ×40).

**Table 1 ijms-23-09125-t001:** The patient characteristics.

	Control *n* = 4	Asthma *n* = 4	COPD *n* = 4	Overall *p*-Value
Age (years)	36 (27–44.5)	61 (38–71)	67 (62–72.5)	0.06
Gender (F/M)	4/0	1/3	2/2	0.09
BMI (kg/m^2^)	22.4 (20.3–23.1)	27.2 (26–30.1)	28 (25.8–30.9)	0.025 *
Atopy (*n*)	2	3	0	0.03
Smoking exposure (pack-years)	0 (0–3.5)	0 (0–0.75)	25 (20–52)	0.015 *
FEV_1_ (% predicted)	105.5 (101–109.5)	84 (81–100)	53 (47–61)	0.018 *
FEV_1_/VC (%)	100.5 (98.5–106.5)	76.3 (70–80.8)	53 (47–61)	0.013 *
FeNO (ppb)	9.3 (9.3–9.3)	47.5 (29.6–67.7)	22.4 (13.9–37.3)	0.124
ACT (points)	N.A.	19 (10–22)	N.A.	N.A.
ICS treatment (*n*)	N.A.	2	0	N.A.
CAT (points)	N.A.	N.A.	11 (8–17)	N.A.
mMRC (points)	N.A.	N.A.	3 (1–3)	N.A.

Data are presented as median (IQR) or *n*. BMI—body mass index, FEV_1_—forced expiratory volume at first second, VC—vital capacity, FeNO—fractional exhaled nitric oxide, ACT—asthma control test, ICS- inhaled corticosteroids, CAT—COPD assessment test, mMRC—modified Medical Research Council, N.A.—not applicable. * control vs. COPD.

## Data Availability

The datasets generated and/or analyzed during the current study are available in the Gene Expression Omnibus repository, accession number GSE175541. https://www.ncbi.nlm.nih.gov/geo/query/acc.cgi?acc=GSE175541, accessed on 6 August 2022.

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
