# Peer review of "RNA-Seq Analysis of UPM-Exposed Epithelium Co-Cultivated with Macrophages and Dendritic Cells in Obstructive Lung Diseases"

_ijms, 2022, doi:10.3390/ijms23169125_

Round 1
Reviewer 1 Report
The manuscript is a very promising and interesting approach that aim at compensating the lack of mechanistic understanding of epithelial dysfunction occurring in muco-obstructive lung diseases and is a pioneeristic attempt to characterize the complex interconnection among immune system and epithelium in response to toxic environmental triggers.
Despite the approach is intriguing a deep and systematic characterization of the epithelium differentiation upon the 21 day of ALI culture is lacking. It is fundamental to understand the level of cell differentiation obtained and the frequency of the different cell types. Filling this aspect will make available and shareable the knowledge generated by the authors.
1. Figure 1. the cartoon is useful to help reader. But to better support the reader understanding the scheme need to be more clear and the caption more detailed to describe every aspect of the figure (acronyms, number, arrows and so on).
2. Figure 2. Do you have correspondence of the identified genesets with other reported datasets?
3. RNA-seq analysis has been performed fresh at the time of nasal brush or upon the 21 days of cell culturing in ALI conditions?
4. Figure 3. Pathway analysis and gene anthology are extremely interesting, although more detailed procedure for method of analysis need to be described and proper statistical method applied. P-value only is not sufficient; you need to compute FDR- adjusted probability instead.
5. A deep and systematic characterization of the epithelium differentiation is lacking. It is fundamental to understand the level of cell differentiation obtained and the frequency of the different cell types. This point is crucial. The author needs to provide evidence for cilia formation, tight junction creation, and define the frequency for every epithelial cell type upon the 21 day of ALI culturing.
6. A similar characterization should be done also for the effect of immune cell co-culture with a characterization of the expressed cytokines to be matched with the RNA-seq analysis.
Reviewer 2 Report
The report contains interesting results that hopefully would be confirmed in a larger group of patients.
Authors should better present, and discuss, how they selected genes that were investigated by rt-PCR (i.e. genes that remained significant after correction for multiple comparisons) and why some of them were upregulated or downregulated in mono and co-cultures.
Author Response
The report contains interesting results that hopefully would be confirmed in a larger group of patients.
Authors should better present, and discuss, how they selected genes that were investigated by rt-PCR (i.e. genes that remained significant after correction for multiple comparisons) and why some of them were upregulated or downregulated in mono and co-cultures. – Fig 5
We want to thank Reviewer for this important suggestion. The selection procedure of genes designed for verification by rt-PCR has been expanded (section 2.3 RT-qPCR analysis). Also Fig. 5 was modified to be more readable. We hope that this version is more understandable for the Reader.
The following fragment has been added. “Genes with p-value (corrected) lower than 0.1 in comparison between UPM exposed and nonexposed epithelial cells from monoculture or triple co-culture in control, asthma and COPD group were selected for verification. Differences in expression observed in RNA-Seq for selected genes are illustrated on Fig. 5 as following:
- Fold change of mRNA expression between UPM exposed and nonexposed epithelial cells from triple co-culture (Fig. 5a, Y-axis) was compared to the fold change of mRNA expression between UPM and no UPM exposed epithelial cells from monoculture (Fig. 5a, X-axis). Separate plot (and gene selection) was prepared for each group (control/asthma/COPD).
- Fold change of mRNA expression between UPM exposed and nonexposed epithelial cells from triple co-culture in one of clinical group was plotted against the same value in other clinical group (three panels: asthma-control, COPD-control, COPD-asthma).”